# Material Characteristics and Comparison of Silver Foil Glass Beads Excavated from the Tomb of King Muryeong in Korea

**Eun A Kim and Gyu Ho Kim ***

Department of Cultural Heritage Conservation Science, Kongju National University, Gongju 32588, Korea; kea0702@smail.kongju.ac.kr
* Correspondence: kimgh@kongju.ac.kr

**Abstract:** This study investigated the comprehensive characteristics of silver foil beads excavated from the Tomb of King Muryeong, based on composition analysis of the foil and glass and morphological characterization. The major element components of metal foil and glass were investigated using a scanning electron microscope (SEM) equipped with an Energy Dispersive X-ray spectrometer (EDS). Trace elements were determined using LA-ICP-MS to constrain the source of the raw material. The morphological characteristics of the beads were recognized through an optical microscope. As a result of the analysis, the metal foil was detected as pure in both gold- and silver-colored glass beads. The chemical composition of the glass was determined as a soda glass, with $Na^+$ acting as a flux and CaO added as a stabilizer to improve durability. It was confirmed that plant ash was used because MgO and $K_2O$ were found to contain more than 1.5%. Through trace element analysis, it was confirmed that the gold and silver-colored silver foil glass beads were made of plant ash glass using different materials. The content of $Fe_2O_3$ was significantly higher in the gold-colored silver foil glass beads than in the silver-colored silver foil glass beads. Therefore, it can be interpreted that the gold-colored silver foil glass beads excavated from the tomb of King Muryeong intentionally achieved the appearance of gold foil glass beads by controlling the color of the outer glass. The silver foil glass beads showed morphological differences according to the color of the outer glass. The gold-colored silver foil glass beads were manufactured as single or segment types, but the silver-colored silver foil glass beads were manufactured as segment types.

**Keywords:** silver foil glass beads; gold foil glass beads; soda; plant ash; archaeological chemistry

## 1. Introduction

Ancient Korean glass beads have been found in various forms, such as round beads, carved beads, tubular beads, and segment beads. Among them, the most extraordinary beads are gold and silver foil glass beads. These glass beads are manufactured by wrapping a primarily prepared inner glass with metal foil and then wrapping it with the outer glass. These beads are classified into gold and silver foil beads depending on the materials of the metal foil. These beads have been given various names by different researchers, including "gold foil glass beads or silver foil glass beads" [1], which are classified according to the materials of the metal foil through composition analysis, and "gold foil segment type beads" [2] and "double-layer glass beads" [3], which are categorized according to the major shapes of the beads. Researchers from countries other than South Korea have utilized terms such as "gilt glass beads" or "gold-in glass beads" [4] and "gold glass beads" [5] to describe the materials.

Gold foil beads in Ancient Korea were created before silver foil beads, and the oldest ones were identified in excavated artifacts from the Hakgok-ri ruins in Yeoncheon, which were constructed in the second century AD. These artifacts were dated to the seventh century AD. There were 38 excavation locations across the Korean Peninsula, and these sites were concentrated in coastal areas, mainly on the west coast, where Baekje was located,

and the southern coast, where Silla and Gaya were located. Silver foil beads have been found in a large quantity only in the Tomb of King Muryeong, Gongju, the second capital of Baekje, which reveals that they were associated with higher social status in the sixth century AD.

Previous studies regarding metal foil glass beads in South Korea have been conducted mainly in the field of conservation science. The earliest research achievements were in the analysis of beads excavated from the Ancient Tombs of Yangdong-ri, Gimhae, which date back to the late third century AD, and from the Naju Bokam-ni third tumulus, which dates back between the sixth and seventh centuries AD [6]. This analysis revealed that the metal foil was made of gold, and the glass was soda-lime glass, because it had a CaO content of 5% or more. Afterward, scientific analysis was conducted on gold foil beads excavated from remains in Myeongam-ri Bakjimeure, Asan [7], and Seokchon-dong, Seoul [8].

As shown in the above literature review, research in South Korea has focused mainly on gold foil beads, and there has been no analysis of glass beads excavated from the Tomb of King Muryeong in Gongju, where silver foil beads were found. The purpose of this study was to analyze the characteristics of manufacturing techniques by identifying the material interpretation and morphological characteristics of the silver foil beads of the tomb of King Muryeong of Gongju, which have an absolute age, through composition analysis of the metal foil and glass.

## 2. Research Target

The Tomb of King Muryeong, which is located in Gongju, Chungcheongnam-do, was built in the Ungjin Period of Baekje (Figure 1).

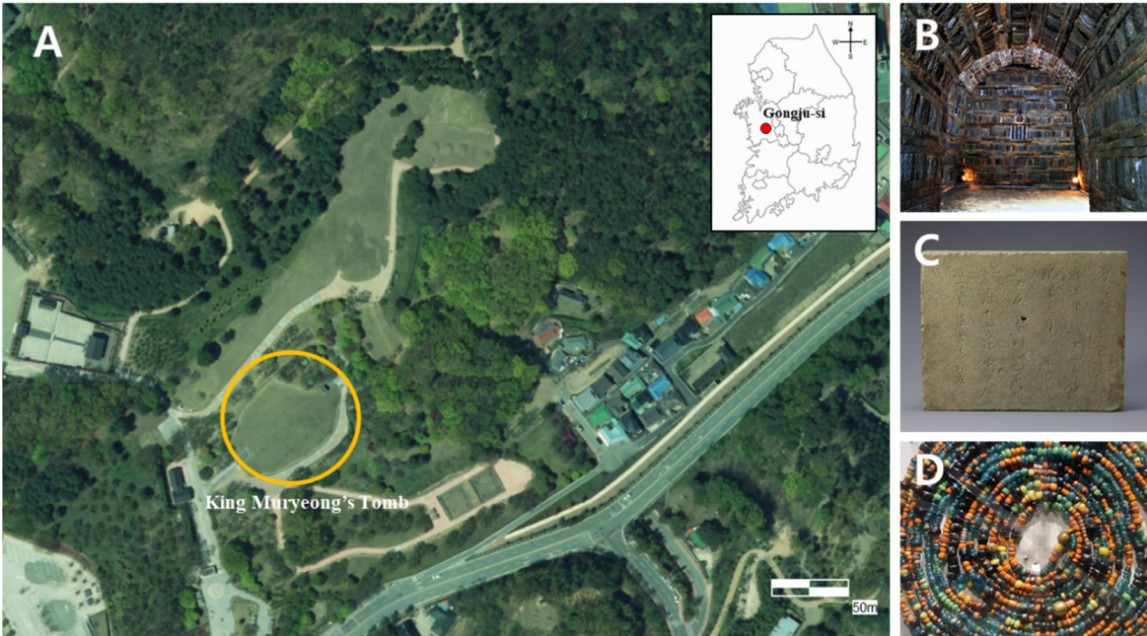

**Figure 1.** Location of the Tomb of King Muryeong selected for research; (**A**): Location of the ruins, (**B**): Brick tomb, (**C**): Memorial tablets with the date of creation of the cemetery, (**D**): Various colors of glass beads.

Among the seven tombs presumed to be the royal tombs of Baekje, the Tomb of King Muryeong is the only royal brick tomb in South Korea, and it was identified because its buried memorial tablets indicated the period in which it was constructed. The value of this tomb, which was located in the ancient capital of Baekje, was immense, and it is registered as a Baekje Historic Area in the list of UNESCO World Heritage Sites [9]. The Tomb of King Muryeong was excavated on 5 July 1971, in the locations where the king and queen were buried. King Muryeong was the 25th king of Baekje, and he sought to stabilize the country

during the period of turmoil in which Goguryeo conquered part of his territory and to revitalize the country through diplomatic relations with the Chinese Southern Dynasties, the Yang and Wa dynasties, in response to Goguryeo advancing to the south. Numerous decorative relics representing the Baekje era were identified in the Tomb of King Muryeong. The records of the buried memorial tablets revealed that King Muryeong died in May 523 and was enshrined in August 525, whereas the queen died in November 526 and was enshrined in February 529 [10]. Among the tomb's artifacts, 171 pieces of gold foil and silver foil beads were found on the queen's chest. In addition, numerous glass beads were found around the king and queen, which suggests that they were used for various decorative purposes.

The 171 gold and silver foil beads excavated from the Tomb of King Muryeong were divided into 122 gold foil and 49 silver foil beads. The gold foil beads were further subdivided into 65 single-type beads and 57 segment-type beads; all silver foil beads were segment-type beads (Figure 2). They were analyzed in six samples, which were classified into three gold foil beads (mr-13~15) and three silver foil beads (mr-16~18).

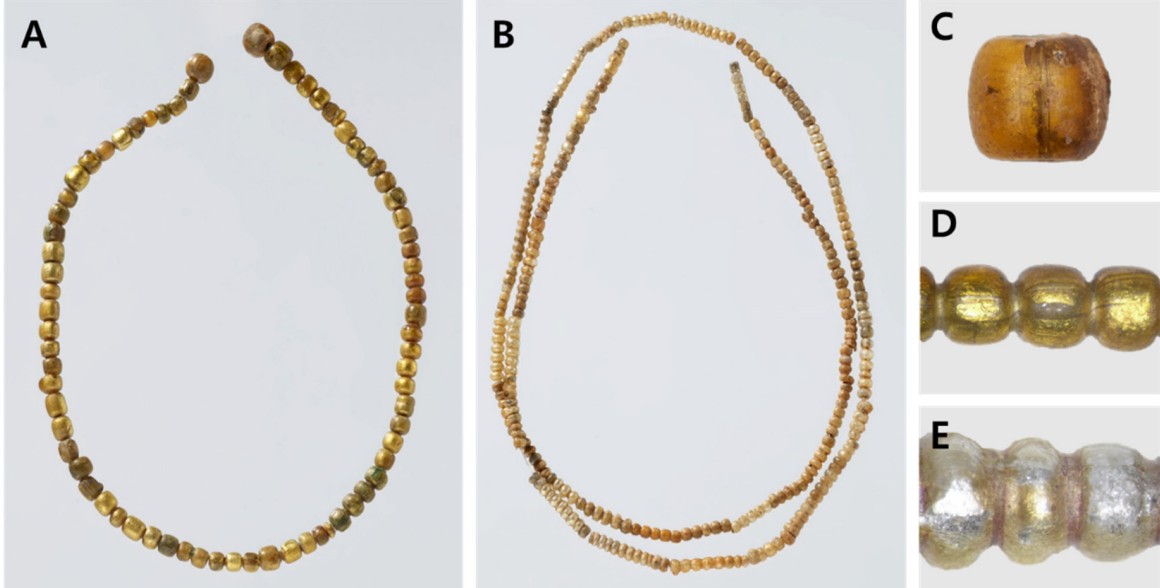

**Figure 2.** Shape of gold and silver foil glass beads excavated from the tomb of King Muryeong in Korea; (**A**): Gold foil glass beads, (**B**): Sliver foil glass beads, (**C**): Gold foil glass single type, (**D**): Gold foil glass segment type, (**E**): Sliver foil glass segment type.

## 3. Methods

### 3.1. Major Element Composition Analysis

The analysis of metal foil and glass was performed by utilizing an energy dispersive X-ray spectrometer (EDS, Oxford, INCA x-sight) attached to an electron microscope (SEM, JEOL JSM-5400). The samples were washed with demineralized water, and then their cross-sections were fixed with an epoxy resin, after which they were polished with sandpaper (Nos. 400, 600, 800, and 1200) followed by an abrasive (6 μm and 1 μm). At each polishing step, the samples were washed three times for 5 min with an ultrasonic cleaner to prevent them from becoming contaminated, and the conductivity of the samples was imparted and analyzed while minimizing the effect on the composition ratio through carbon coating. Before analysis, reproducibility by electron beam emission was verified with 99.9% cobalt for correction, and measurements were obtained at 150-s intervals under an acceleration voltage of 20 kV and a measurement distance of 20 mm. The metal foil was measured by spot and analyzed. Glass was analyzed at 2000 magnification using a scanning electron microscope by dividing the inner and the outer glass. Measurement results were normalized to 100%. Data less than 0.1% were treated as ND (Not Detected). The reliability of the

compositional analysis was tested after five measurements of a glass standard sample (SRM 620) compared to the properties of the original composition (Table 1).

**Table 1.** Composition of the glass standard SRM 620.

| Standard Sample | Section | Chemical Composition (wt.%) | | | | | | | | Total |
|---|---|---|---|---|---|---|---|---|---|---|
| | | SiO$_2$ | Na$_2$O | K$_2$O | CaO | Al$_2$O$_3$ | MgO | TiO$_2$ | Fe$_2$O$_3$ | |
| SRM 620 | Certified Value | 72.08 | 14.39 | 0.41 | 7.11 | 1.80 | 3.69 | 0.09 | 0.04 | 99.61 |
| | Measured Value | 72.26 | 14.38 | 0.43 | 7.08 | 1.95 | 3.83 | 0.03 | 0.04 | 100.0 |
| | | *0.23* | *0.18* | *0.01* | *0.09* | *0.09* | *0.18* | *0.01* | *0.01* | |

### 3.2. Trace Element Analysis

Laser Ablation-Inductively Coupled Plasma-Mass Spectroscopy (LA-ICP-MS, Thermo Fisher Scientific Inc. ELEMENT XR with VG UV Laser Probe) was used for trace element analysis to determine the source of the raw materials used in the glass. The laser system uses Nd; YAG to generate a laser with a wavelength of 266 nm in the UV region, and it was specified at a maximum of 15 Hz, and the maximum energy was performed at 3–4 mJ. Erosion caused by the laser occurs at a diameter of 60–10 μm and a depth of 250 μm, depending on the repetition rate and duration.

### 3.3. Morphological Analysis

An optical microscope (Leica M$_Z$75) was used to determine the color of the glass, the state of air bubbles, the degree of cracking and surface weathering, and the manufacturing technique, and an SEM was used to observe the microstructure and impurities, as well as the state and weathering layer of the metal foil inside the glass.

## 4. Results and Discussion

### 4.1. Major Elements Composition

#### 4.1.1. Metal Foil

After observation and analyses of the metal foils of gold and silver foil beads via SEM, quantitative analysis results were summarized as the mean and standard deviation values for each sample (Table 2, Figure 3). Pure silver was detected in the metal foils of both the gold and silver foil beads, which indicates that they were not alloyed with other materials. Deviation in the amount of silver varies considerably depending on the proportions of the detected S, Cl, Br, and O, which are considered to be corrosive compounds [11,12]. Therefore, the gold and silver foil beads excavated from the Tomb of King Muryeong were all interpreted to be de Facto silver foil beads because the metal foils were identified as silver foil.

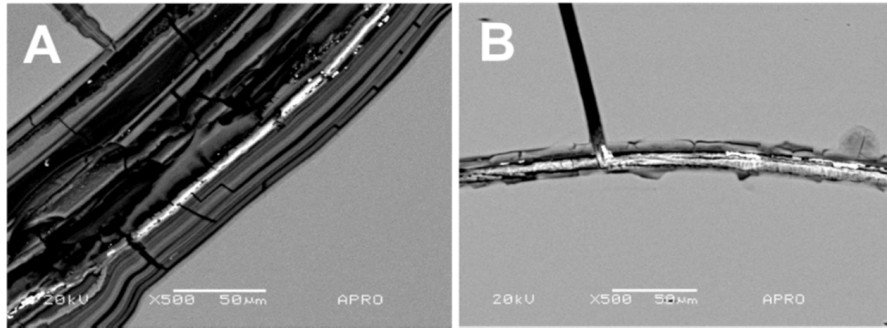

**Figure 3.** SEM images of silver foil glass beads from the tomb of King Muryeong in Korea. (**A**): Gold-colored (mr-13), (**B**): Silver-colored (mr-16).

**Table 2.** SEM-EDS analytical results of metal foil in foil glass beads excavated from the Tomb of King Muryeong in Korea.

| Sample Number | Bead Color | Chemical Composition (wt.%) | | | | | | | | Total |
|---|---|---|---|---|---|---|---|---|---|---|
| | | Ag | Cu | Au | Fe | S | Cl | Br | O | |
| mr-13 | Gold | 64.43 | 0.22 | ND | 0.43 | ND | 11.32 | 13.46 | 10.14 | 100 |
| | | *0.98* | *0.11* | *-* | *0.14* | *-* | *2.15* | *1.52* | *3.62* | *-* |
| mr-14 | Gold | 62.10 | 0.32 | 1.28 | 1.09 | ND | 2.34 | 4.38 | 28.49 | 100 |
| | | *6.40* | *0.26* | *0.46* | *0.11* | *-* | *1.93* | *2.08* | *2.16* | *-* |
| mr-15 | Gold | 67.90 | 0.30 | ND | 0.49 | ND | 5.79 | 22.30 | 3.22 | 100 |
| | | *8.50* | *0.09* | *-* | *0.06* | *-* | *1.45* | *9.02* | *0.58* | *-* |
| mr-16 | Sliver | 63.60 | 0.11 | ND | 0.37 | ND | 7.07 | 4.93 | 23.92 | 100 |
| | | *4.98* | *0.13* | *-* | *0.14* | *-* | *0.26* | *1.45* | *3.91* | *-* |
| mr-17 | Sliver | 73.54 | ND | ND | 0.36 | ND | 6.19 | 4.05 | 15.77 | 100 |
| | | *8.62* | *-* | *-* | *0.10* | *-* | *0.74* | *1.61* | *6.51* | *-* |
| mr-18 | Sliver | 74.74 | 0.14 | 0.57 | 0.85 | 10.50 | ND | 1.85 | 11.33 | 100 |
| | | *7.37* | *0.15* | *1.02* | *0.96* | *0.63* | *-* | *1.28* | *6.65* | *-* |

ND: Not Detected.

### 4.1.2. Glass

The chemical composition of six silver foil glass beads divided into three gold-colored and three silver-colored glass beads was analyzed (Table 3).

### 4.2. Trace Element Composition

According to the analysis results, the gold-colored and silver-colored glass beads were relatively similar in their main composition, excluding the abundance of chemical elements correlated to the colorants of the glass, and the inner and outer glass surrounding the metal foil was identified as being made of the same components. The flux, stabilizer, characteristics of the raw materials, and colorants were referred to in the composition classification of glass excavated in south Korea [13–15].

Regarding flux characteristics, the $Na_2O$ content ranged from 16.6 to 18.5% on average, indicating that the beads were made of soda glass. The CaO content was high, in the range of 5.2 to 7.2% for the stabilizer component, which was identified as an HCLA (CaO > 5%, $Al_2O_3$ < 5%). It was measured to be 2.60–3.26% MgO and 2.96–4.14% K2O. Therefore, it is identified as HMK type, plant ash glasses. Gold- and silver-colored beads were further classified depending on the colorant, such as TiO2, MnO, $Fe_2O_3$, and CuO, which are the main color pigments of glass. The gold-colored beads had a high $Fe_2O_3$ content, ranging from 4.9 to 5.8%, which indicates that the silver foil beads appeared amber in color. Thus, the gold-colored beads were silver foil beads, as their inner metal foil was identified as silver through the composition analysis, whereas their outer and inner glasses were fabricated to be amber in color by using $Fe_2O_3$ to give them the appearance of gold foil beads. The silver foil beads' low $Fe_2O_3$ content of 0.2% was manufactured so that the inner silver foil was visible, because their outer and inner glass were colorless (Figure 4A). Although gold-colored silver foil glass beads and silver-colored silver foil glass beads were identified as a glass of soda-plant ash, the principal component statistical analysis was performed to determine whether the glass was made using a similar material.

The silver foil glass beads of King Muryeong were classified into three types (Figure 4B). Type I was gold-colored silver foil beads (mr-13, 15) that had an amber glass color. Type II corresponded to silver-colored silver foil beads (mr-16, 17) that were colorless glass. Type III consisted of a gold-colored (mr-14) and a silver-colored (mr-18) silver foil glass bead. Therefore, it was confirmed that the silver foil glass beads of King Muryeong were of the same flux and stabilizer, but the gold-colored and silver-colored silver foil beads were manufactured using different raw materials.

**Table 3.** SEM-EDS analytical results of the glass layer of silver foil glass beads excavated from the Tomb of King Muryeong in Korea.

| Sample Number | Color | | Layer | Chemical Composition (wt.%) | | | | | | | | | | | | | Total | Remark |
| | Bead | Glass | | $SiO_2$ | $Na_2O$ | $K_2O$ | CaO | $Al_2O_3$ | MgO | $TiO_2$ | MnO | $Fe_2O_3$ | CuO | PbO | $SO_3$ | Cl | | |
| mr-13 | Gold | Amber | Outer | 59.77 | 18.32 | 2.98 | 5.94 | 2.29 | 2.64 | 0.13 | 0.30 | 6.13 | nd | nd | 0.21 | 1.28 | 100 | - |
| | | | | *0.43* | *0.18* | *0.02* | *0.05* | *0.06* | *0.07* | *0.06* | *0.04* | *0.17* | *-* | *-* | *0.09* | *0.04* | *-* | |
| | | | Inner | 61.92 | 18.19 | 2.99 | 5.37 | 2.23 | 2.60 | 0.15 | 0.19 | 4.75 | nd | 0.12 | 0.28 | 1.20 | 100 | - |
| | | | | *0.68* | *0.17* | *0.06* | *0.08* | *0.06* | *0.09* | *0.04* | *0.02* | *0.17* | *-* | *0.09* | *0.04* | *0.06* | *-* | |
| mr-14 | Gold | Amber | Outer | 57.53 | 18.60 | 4.14 | 5.47 | 3.54 | 3.26 | 0.23 | 0.15 | 5.77 | nd | nd | 0.33 | 1.00 | 100 | - |
| | | | | *0.27* | *0.25* | *0.03* | *0.08* | *0.08* | *0.08* | *0.04* | *0.04* | *0.12* | *0.05* | *-* | *0.05* | *0.05* | *-* | |
| | | | Inner | 59.14 | 17.60 | 3.95 | 4.86 | 4.15 | 3.05 | 0.30 | 0.15 | 5.55 | nd | nd | 0.34 | 0.91 | 100 | - |
| | | | | *0.43* | *0.06* | *0.07* | *0.07* | *0.12* | *0.08* | *0.06* | *0.04* | *0.13* | *-* | *-* | *0.13* | *0.03* | *-* | |
| mr-15 | Gold | Amber | Outer | 59.95 | 18.68 | 2.96 | 5.94 | 2.26 | 2.68 | 0.12 | 0.30 | 5.54 | nd | nd | 0.26 | 1.30 | 100 | Weathering |
| | | | | *0.28* | *0.09* | *0.02* | *0.09* | *0.12* | *0.07* | *0.01* | *0.04* | *0.11* | *-* | *-* | *0.07* | *0.06* | *-* | |
| | | | Inner | 62.75 | 18.15 | 2.99 | 5.36 | 2.13 | 2.54 | 0.15 | 0.24 | 4.23 | nd | nd | 0.29 | 1.18 | 100 | Weathering |
| | | | | *0.53* | *0.41* | *0.07* | *0.20* | *0.11* | *0.11* | *0.04* | *0.05* | *0.76* | *-* | *-* | *0.08* | *0.06* | *-* | |
| mr-16 | Sliver | Colorless | Outer | 62.30 | 16.82 | 3.07 | 7.26 | 5.18 | 2.70 | 0.20 | nd | 1.25 | nd | nd | 0.46 | 0.77 | 100 | - |
| | | | | *0.80* | *0.19* | *0.05* | *0.03* | *0.06* | *0.06* | *0.09* | *-* | *0.09* | *-* | *-* | *0.08* | *0.06* | *-* | |
| | | | Inner | 62.07 | 16.89 | 3.10 | 7.37 | 5.05 | 2.73 | 0.21 | nd | 1.18 | 0.11 | nd | 0.50 | 0.79 | 100 | - |
| | | | | *0.24* | *0.24* | *0.03* | *0.15* | *0.08* | *0.11* | *0.04* | *-* | *0.07* | *0.10* | *-* | *0.07* | *0.04* | *-* | |
| mr-17 | Sliver | Colorless | Outer | 62.38 | 16.71 | 3.09 | 7.35 | 5.08 | 2.74 | 0.21 | nd | 1.23 | nd | nd | 0.43 | 0.78 | 100 | - |
| | | | | *0.48* | *0.11* | *0.07* | *0.12* | *0.12* | *0.11* | *0.03* | *-* | *0.04* | *-* | *-* | *0.05* | *0.06* | *-* | |
| | | | Inner | 61.69 | 17.73 | 3.04 | 7.17 | 5.01 | 2.77 | 0.26 | nd | 1.11 | nd | nd | 0.43 | 0.80 | 100 | - |
| | | | | *0.33* | *0.18* | *0.06* | *0.13* | *0.07* | *0.05* | *0.06* | *-* | *0.09* | *-* | *-* | *0.08* | *0.05* | *-* | |
| mr-18 | Sliver | Colorless | Inner | 61.08 | 18.77 | 4.03 | 5.86 | 4.34 | 2.99 | 0.19 | nd | 1.22 | nd | 0.12 | 0.29 | 1.10 | 100 | Outer(X) |
| | | | | *0.34* | *0.25* | *0.11* | *0.07* | *0.16* | *0.06* | *0.04* | *-* | *0.13* | *-* | *0.10* | *0.07* | *0.02* | *-* | |

nd: Not Detected.

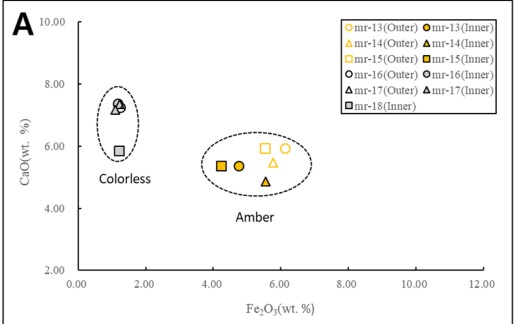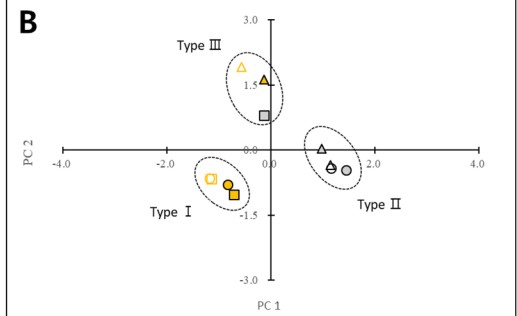

**Figure 4.** Composition comparison of silver foil glass beads; (**A**): $Fe_2O_3$ vs. CaO, (**B**): Principal component analysis.

The silver foil beads found in the Tomb of King Muryeong in Gongju exhibited contrasting characteristics to the gold foil beads found on the Korean Peninsula. Most gold foil beads have been identified as Roman glass, which is Natron glass, while silver foil beads are plant ash glass, which is a type of Sasanian glass that was popular in Europe and Mesopotamia [16]. Moreover, gold-colored silver foil beads excavated from the Bukel ruins in Albania, an Eastern European country, with the same period of construction as the Tomb of King Muryeong in Gongju, were also identified as plant ash glass [17]. These results suggest that silver foil beads are foreign relics that were imported through trade rather than manufactured on the Korean Peninsula. Thus, once enough data is accumulated to determine whether these are related to the origin of the ancient silver foil beads, further studies are necessary to determine the correlation in terms of materials.

The outer glass of the silver foil beads was analyzed by LA-ICP-MS to obtain quantitative values for 20 trace elements (Table 4). In the trace elements, there was a high quantity of phosphorus in the silver foil glass beads, which is closely related to plant ash material [18]. The gold-colored and silver-colored silver foil beads were different in their Sr, Zr, Ce, and Rh contents. This composition of the gold-colored silver foil glass was detected as being higher. It is presumed that this difference is due to the difference in the raw material used to make the glass: sand [19].

**Table 4.** LA-ICP-MS analytical results of silver foil glass beads excavated from the tomb of King Muryeong.

| Element (ppm) | Gold-Color Beads (Amber Glass) | | | Silver-Color Beads (Colorless Glass) | | |
|---|---|---|---|---|---|---|
| | mr-13 | mr-14 | mr-15 | mr-16 | mr-17 | mr-18 |
| Co | 5 | 6 | 5 | 4 | 5 | 3 |
| Zn | 25 | 54 | 38 | 54 | 50 | 41 |
| P | 5044 | 5597 | 5477 | 4384 | 4555 | 5162 |
| Ba | 147 | 169 | 144 | 260 | 244 | 193 |
| B | 140 | 198 | 154 | 191 | 213 | 171 |
| V | 20 | 35 | 20 | 24 | 25 | 18 |
| Cr | 14 | 16 | 15 | 18 | 18 | 9 |
| Ni | 12 | 16 | 13 | 12 | 12 | 8 |
| As | 5 | 6 | 6 | 2 | 2 | 1 |
| Rb | 27 | 37 | 29 | 65 | 60 | 44 |
| Sr | 281 | 805 | 278 | 1146 | 1116 | 542 |
| Ce | 21 | 27 | 22 | 46 | 46 | 42 |
| Y | 14 | 10 | 12 | 16 | 15 | 13 |
| Zr | 75 | 89 | 64 | 137 | 120 | 114 |
| Nb | 3 | 4 | 3 | 11 | 11 | 11 |
| La | 13 | 15 | 12 | 22 | 21 | 18 |
| Nd | 10 | 12 | 10 | 19 | 18 | 17 |
| Th | 4 | 5 | 3 | 8 | 8 | 5 |
| U | 1 | 2 | 1 | 2 | 2 | 1 |
| Li | 25 | 50 | 25 | 38 | 33 | 29 |

*4.3. Morphological Characteristics*

According to the observations of the cross-sections of gold- and silver-colored beads through an optical microscope, the outer glass of the gold-colored beads was amber, whereas the outer glass of the silver-colored beads was colorless (Figure 5). There were also morphological differences in the cross-sections of the gold and silver-colored beads. As shown in Figure 5A, traces of thermal processing were observed in the gold-colored beads with round outer and inner glasses after they were manufactured to ensure metal foil bonding. These reveal that a perforation was formed inside the inner glass. Silver foil beads were divided into two types according to their shape. Type I had a perforation on the cross-section of the inner glass-like gold-colored glass, as shown in Figure 5B, whereas Type II had an inner glass with an optical linear shape similar to a long rod or bamboo, as shown in Figure 5C. In the case of Type II, a long-shaped inner glass was prepared and covered with metal foil, after which the boundary between the beads was created by bonding the outer glass, only without subsequent heat processing. This characteristic was identified only in the silver foil beads.

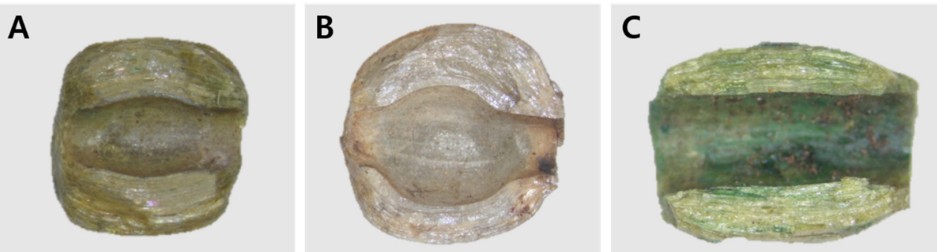

**Figure 5.** Morphological characteristics of silver foil beads from the tomb of King Muryeong in Korea; (**A**): Gold-colored, (**B**): Silver-colored type I, (**C**): Silver-colored type II.

**5. Conclusions**

This study investigated the comprehensive characteristics of silver foil beads excavated from the Tomb of King Muryeong in South Korea based on composition analysis and morphological characterization of the metal foil and glass.

Both the gold and silver foil beads excavated from the Tomb of King Muryeong in Gongju were found to contain pure silver in their metal foil. Thus, they were all determined to be silver foil glass beads. The silver foils were divided into gold-colored and silver-colored beads. The results of composition analysis on the gold-colored and silver-colored glass beads revealed that they were made of soda glass, because they had $Na_2O$ as a flux and $CaO$ as a stabilizer, and they were identified as plant ash glass based on their $K_2O$ and $MgO$ contents. It was demonstrated that the gold-colored and silver-colored glass beads were produced using different raw materials. This is also confirmed in Sr, Zr, Ce, and Rh, which could suggest the origin of characteristic elements of sand.

The gold foil glass beads that have been identified in South Korea to date were all Natron glass, showing differences in chemical composition from the silver foil glass beads, which were plant ash glass. Plant ash glass has a chemical composition similar to that of Sasanian glass, which was prevalent in Mesopotamia, and they are therefore considered to have been imported through trade rather than manufactured in the Korean Peninsula. However, plant ash glass is not a common component of glass beads. For this reason, further studies are necessary to clarify the correlation in terms of raw materials. The colors of the beads were identified based on the $Fe_2O_3$ content. The $Fe_2O_3$ was high in gold-colored beads, whereas the outer and inner glasses were amber. In contrast, the $Fe_2O_3$ content was low in the silver foil glass beads, where the outer and inner glasses were colorless. Thus, the silver foil glass beads from the Tomb of King Muryeong suggested that the effects of silver and gold foil were deliberately employed via adjustment of the colorant of the glass. The morphological characteristics varied depending on whether the beads were gold or silver foil glass. The gold-colored glass beads had perforations inside, which

are traces of heat treatment performed to create a smooth bond between the metal foil and the glass. Silver-colored beads were divided into two types: beads with perforations and beads with bamboo-shaped straight inner glass with only the outer glass being heat-treated.

**Author Contributions:** Conceptualization, G.H.K. and E.A.K.; methodology, G.H.K.; software, E.A.K.; validation, G.H.K. and E.A.K.; formal analysis, G.H.K. and E.A.K.; investigation, G.H.K. and E.A.K.; resources, G.H.K.; data curation, G.H.K. and E.A.K.; writing—original draft preparation, G.H.K. and E.A.K.; writing—review and editing, E.A.K.; visualization, E.A.K.; supervision, G.H.K.; project administration G.H.K.; funding acquisition, G.H.K. All authors have read and agreed to the published version of the manuscript.

**Funding:** This research was supported by Basic Science Research Program through the National Research Foundation of Korea (NRF), funded by the Ministry of Education (NRF-2021R1I1A2040339) and was supported by a research grant from Kongju National University in 2021.

**Institutional Review Board Statement:** Not applicable.

**Informed Consent Statement:** Not applicable.

**Data Availability Statement:** Not applicable.

**Acknowledgments:** We would like to thank James W. Lankton, who measured LA-ICP-MS in this paper and assisted in the study of ancient glass beads in Korea.

**Conflicts of Interest:** The authors declare no conflict of interest.

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
