# Peer review of "Material Characteristics and Comparison of Silver Foil Glass Beads Excavated from the Tomb of King Muryeong in Korea"

_applsci, doi:10.3390/app12136385_

Round 1
Reviewer 1 Report
This is the third time I revised this paper, the first before authors re-submisison.
I found some improvements in data presentation (tables are better presented for example) and discussion with also a better focus in the conclusion paragraph but I believe that authors can still do something more to make the paper more meaningful. The object of the study is interesting but the discussion is still poorly expanded and some (new) data (i.e. trace elements) are just introduced without a real argumentation. There are many aspect that can be considered together to improve this part and make the paper a good publication, as I annotated in the attached pdf.
Two important things:
1) the authors did not perform any check by EPMA analyses of the data obtained on glasses (that they carried out by SEM) as I've already suggested since my first revision, even worse they totally omitted to reply to this comment in their cover letter.
2) again, in my first revision I asked why the analyses on Corning standards were so bad. The authors again totally ignored this comment and simply changed the reference standard to SRM620...I would like an explanation for that.
Finally, even if the English style have been improved from the first submission, I think that in some parts it still should be revised, especially in the abstract.
All the others specific comments are annotated in the attached pdf.
In conclusion, I think that the finding of the presented study are nice and suitable for publication but the authors should do some more efforts especially in the data discussion. I chose minor revision but it probably best the moderate option. However, my hope is that these comments are helpful to the authors. They are intended to be constructive with the goal of allowing the data and the ideas to reach publication.

Author Response
Thank you for reviewing our paper three times.
This thesis is my first submission in English. As this is my first thesis, I think there are many shortcomings. Therefore, I hope that your kind advice and help will be of great help in completing my thesis. I may not be able to communicate much due to my lack of English, but I will answer the review as follows.
Answers to important points pointed out in the thesis review:
- We also think it is good to perform with EPMA in the glass analysis mentioned by the reviewer. However, I think that useful information can be obtained from SEM-EDS if the measurement conditions are well established for the major component of glass. And all glass analysis analyzed in Korea so far is the result of SEM-EDS measurement. And we regularly compare accuracy and reproducibility with a glass standard sample (SRM 620).
- The results of the analysis of the Corning standard sample in the first submitted paper were mistakenly published as materials still under review with the corresponding author. Although our laboratory is reviewing the measurement results of Corning standard samples, the composition of soda glass of Corning glass tends to match well, but potassium glass and lead glass have a large error in composition analysis. We are investigating the cause of this, but we have not been able to determine the cause yet. Therefore, in our laboratory, SRM620, which has the best analysis results among standard samples, is used as a standard sample.
We also reflected on the corrections shown in the PDF and responded to comments from reviewers. Please check the attached file. The first thesis written in English seems to be of poor quality, but the reviewer gave generous advice to improve the thesis, so I was able to write a better thesis, and I think it is a great experience for me to grow in the future. Thanks again for the reviewer's kind advice.
Best regards,
Researcher Eun a Kim

Reviewer 2 Report
The paper describes the study of "silver foil" and "gold foil" glass beads, which were produced using a special technique in the 7th c. AD Ancient Korea.
To better understand the production technique of these metal foil glass beads, I would recommend to present a sketchy illustration of the sequential layers of the "outer glass" - "metal foil" - "inner glass" with the characteristic thicknesses.
I would be happy to read a short justification, why it was allowed to perform destructive studies on the objects. Were they originally broken, or were they cut for the sake of the investigations? Are there any alternative way of non-destructive studies?
Finally, what was the ultimate aim of the study? Just to see if there is a difference between the compositions of the "silver foil" and the "gold foil" glass beads? Was it an aim to associate them with certain workshop(s)?
Author Response
Thank you for considering my article for publication in applied science.
We also appreciate the time and effort you and each of the reviewers have dedicated to providing insightful feedback on ways to strengthen our paper.
- The manufacturing technique of metal foil glass beads was able to be identified in detail in this study, but there are insufficient data to integrate them due to the limited case. We will collect more data for future research.
- The sample used for the investigation were found damaged during the excavation process. And since ancient glass uses alkali metals as a material, the non-destructive analysis continues to be studied.
- The purpose of the study was not clearly stated in the text, so it was modified. Through this study, it was confirmed that the gold foil and silver foil beads excavated from the tomb of King Muryeong were made of silver, and there was a difference in composition depending on the color of the glass. Based on this, we plan to research whether the same workgroup exists in Korea.
Thanks again for the reviewer's kind advice.
Best regards,
Researcher Eun a Kim
Reviewer 3 Report
Text must be improved by removing repetitions. The bibliography need to be improved, several studies are reported about metal foil glass beads, not mentioned. The morphological characteristics section is too concise and can be improved. A typos is in the caption of figure 2.
Author Response
Thank you for considering my article for publication in applied science.
We also appreciate the time and effort you and each of the reviewers have dedicated to providing insightful feedback on ways to strengthen our paper.
We actively accepted the points pointed out by the reviewer and corrected them. Thanks again for the good point.
Best regards,
Researcher Eun a Kim
Round 2
Reviewer 1 Report
The authors welcomed many of the comments regarding text style and minor critical aspects, however they still overlooked to answer directly to the two critical points I listed in my cover letter. The authors also believe that it is not necessary to expand or improve the discussion as I suggested. The manuscript, in this form, is sufficiently good for publication but I left to the editor the final decision on that.
Author Response
Thanks for the kind review.
I believe that this paper will serve as an opportunity for better research in the future.
It seems that your comments have helped a lot in improving the thesis.
Thank you again for reviewing our paper.
Reviewer 2 Report
I accept the publication in the present form.
Author Response
Thanks for the kind review.
It seems that your comments have helped a lot in improving the thesis.
Thank you again for reviewing our paper.
This manuscript is a resubmission of an earlier submission. The following is a list of the peer review reports and author responses from that submission.
Round 1
Reviewer 1 Report
The paper “Interpretation of the material characteristics and origins of ancient gold and silver foil glass beads excavated from the Tomb of King Muryeong in Korea” by Kim Eun and Kim Gyu Ho present new, original data on a selection of ancient silver and gold foil glass beads found in a royal tomb in South Korea.
The authors perform morphological observation and chemical analyses in order to characterize and classify the artifacts. From the obtained results they find out that the two different foil glass beads (golda and silver) are instead made by an only Ag-rich foil. They ascribed the golden color to the effect of addition of Fe to the external glass that provide the yellow gold tone to the beads.
The work presented in this paper is interesting and finding are nice, however it suffers of many deficiencies that must be improved before it may be considered suitable for publication.
First of all the English style should be deeply revised also by a native English-speaker. Some sentences are too hard to follow and many terms are used inappropriately. Second and very important: data table are cited but not provided!
Moreover, the principal suggestions is to reconsider, rearrange and expand the data description and results presentation, integrating it with the discussion to improve the significance of the research.
I attached a commented pdf with all the details on major and minor aspects.
In conclusion I suggest a very major revision before publication.

Author Response
Thank you for considering my article for publication in applied science.
We also appreciate the time and effort you and each of the reviewers have dedicated to providing insightful feedback on ways to strengthen our paper.
We have incorporated changes that reflect the detailed suggestions you have graciously provided.
I worked to improve the writing by eliminating any errors in grammar, refining word choice, and improving sentence structure and flow. Tables not attached to the thesis have been re-edited and uploaded. I think there was a mistake in the process of uploading my paper.
Please review the comments throughout for Details of the fix are written in the full comments.
Kind regards,
Kim Eun a

Reviewer 2 Report
Abstract
- It will be clearer for the readers if in the abstract the objects of research are described as gilded and silver-plated with gold and silver foil, glass beads.
- As a summary sentence for this abstract, it is need to mentioned about the origin of glass beads: where they were probably made.
Introduction
- The introduction begins with a description of the different types of glass beads with metal foil according to how they are made, as well as a brief chronology concerning the current interpretation of the appearance of glass beads in the studied region. There are no enough cited literature sources. The introduction should consider where in ancient times such glass beads were produced and it is necessary to cite more literature sources. It is also needed to consider in detail the methods used to examine such artifacts in terms of age, region of production and, if available, data on commercial transfer, and to cite all of them.
Methods
- More than one analytical method is needed to make a detailed hypothesis about the origin of these artifacts. The study of trace elements by ICP LA analysis is a good option. Even if it is not done yet for such artifacts, a start could be made. SEM EDS analysis only is not enough.
If the authors cannot afford to make another analysis showing the trace elements and their quantities, then, at least make a detailed historical analysis of current knowledge about this type of artifacts, the methods by which they are studied, and the analytical data known about such objects
Author Response
Thank you for considering my article for publication in applied science.
We also appreciate the time and effort you and each of the reviewers have dedicated to providing insightful feedback on ways to strengthen our paper.
We have incorporated changes that reflect the detailed suggestions you have graciously provided.
All requests and comments made by the judges have been corrected. However, I would like to inform you that comparison with other literature may lead to biased results in the absence of scientific analysis data, so we reviewed it with our referenced data.
I worked to improve the writing by eliminating any errors in grammar, refining word choice, and improving sentence structure and flow. Tables not attached to the thesis have been re-edited and uploaded. I think there was a mistake in the process of uploading my paper.
Please review the comments throughout for Details of the fix are written in the full comments. Please see the attachment
Kind regards,
Kim Eun a

Round 2
Reviewer 1 Report
Regarding the revision of manuscript applsci-1638201 by Kim Eun A., Kim Gyu Ho I can only appreciate the effort in reviewing the English style as showed by the author in the provided file. Just make sure that the sense of some sentences is maintained.
However, all the other comments regarding analytical aspects as well those regarding interpretation queries and suggestions to improve the discussion and the paper structure have been completely ignored. The attached cover letter is merely the English revised version of the text, without any answer to the comments and critical points made in the revision.
In the revised version of the manuscript, the authors must provide point by point the answer to the reviewer/s comments also discussing the more critical points. Indeed, to one of the more critical point regarding the standard data and the method on SEM-EDS analyses (i.e., Si and Al are far from the certified value, as also Ca and Na in CorningB. How do you manage that? Do you correct the analyses of you data? It is well known that SEM analysis are not fully suitable for accurate glass compositional data. I wold like to see some representative analyses made with EPMA instead of SEM where analyses have a forced closure to 100) the authors just change the reference standard without any comment! Why? An explanation is needed.
Data on samples are now provided but some aspects still need attention: why there is no total? I calculate the totals and the closures are not perfectly 100, so you calibrate your EDS analyses? this is not explained in the method paragraph. Please integrate it. What about EPMA analyses? Why there is oxygen as quantified element in the analyses of the metal foil? which is the sense? Why you do you use the annotation <0.1 in the data table? Better bdl (below detection limit) in any case...moreover what is the benefit of the column named "remark"?
To conclude, in my opinion, the revision of the manuscript is still to be accomplished and the reviewer comments still need to be replied.
All the best.
Author Response
Dear. Reviewer 1
Thank you for considering my article for publication in applied science.
Your great comments have helped a lot in improving the quality of this article. We also hope that our edits and the responses we provide below satisfactorily address all the issues and concerns you and the reviewers have noted.
Here are responses to the reviewer comments:
Commnet 1 Respnses :
The use of standard samples in composition analysis of ancient glass can be an important criterion for the reliability of measurement results. The standard sample generally used in the composition analysis of ancient glass so far is the revised SRM620 in Table 1. And Corning B, C, and D are standard samples that have been used recently, but quantification has not yet been confirmed under the same measurement conditions, and useful measurement conditions are being reviewed. However, I made a mistake in posting the measurement results for standard samples in this submitted article without consulting with my advisor. In this regard, it has been replaced by the measurement results on the SRM 620 standard sample. And so far, our laboratory has been measuring the composition of ancient glass with SEM-EDS. As you mentioned, I agree that EPMA may be more suitable for quantitative analysis than SEM-EDS. However, in our lab, rather than 100% normalization, we are also using a quantitative method that measures glass composition with SEM-EDS and uses standard samples for each element.
Comment 2 Respnses:
In this paper, reflecting the points you pointed out, I would like to organize Tables 1, 2, and 3 into a unified standard as a result of 100% normalization of the measured elements. The reason why oxygen is presented as a quantitative value in Table 2 was not reviewed in this paper, but it is thought that it can be used as a basis for estimating the corrosion degree of silver foil, so it is summarized together. And, although the measurement limit of SEM-EDS was indicated as < 0.1, inconsistent results were confirmed, and this was corrected by unifying it as ND (Not Detected).
Please see the attachment
Again, Proffessor Kim and I sincerely thank you for your comment
Kim Eun a
